# Informing the development of a standardised approach to measure antibiotic use in secondary care: a systematic review protocol

Selina Patel,[1] Arnoupe Jhass,[2] Susan Hopkins,[3] Laura Shallcross[1]

[1]Institute of Health Informatics, University College London, London, UK
[2]Department of Primary Care and Population Health, Institute of Epidemiology and Health Care, University College London, London, UK
[3]Public Health England, London, UK

**Correspondence to**
Selina Patel;
selina.patel.17@ucl.ac.uk

## ABSTRACT

**Introduction** Ecological and individual-level evidence indicates that there is an association between level of antibiotic exposure and the emergence and spread of antibiotic resistance. The Global Point Prevalence Survey in 2015 estimated that 34.4% of hospital inpatients globally received at least one antimicrobial. Antimicrobial stewardship to optimise antibiotic use in secondary care can reduce the high risk of patients acquiring and transmitting drug-resistant infections in this setting. However, differences in the availability of data on antibiotic use in this context make it difficult to develop a consensus of how to comparably monitor antibiotic prescribing patterns across secondary care. This review will aim to document and critically evaluate methods and measures to monitor antibiotic use in secondary care.

**Methods and analysis** We will search Medline (Ovid), Embase (Ovid), Cumulative Index to Nursing and Allied Health Literature, Cochrane Central Register of Controlled Trials and websites of key organisations for published reports where an attempt to measure antibiotic usage among adult inpatients in high-income hospital settings has been made. Two independent reviewers will screen the studies for eligibility, extract data and assess the study quality using the Newcastle-Ottawa scale. A description of the methods and measures used in antibiotic consumption surveillance will be presented. An adaptation of the Affordability, Practicability, Effectiveness, Acceptability, Side-effects Equity framework will be used to consider the practicality of implementing different approaches to measuring antibiotic usage in secondary care settings. A descriptive comparison of definitions and estimates of (in)appropriate antibiotic usage will also be carried out.

**Ethics and dissemination** Ethical approval is not required for this study as no primary data will be collected. The results will be published in relevant peer-reviewed journals and presented at relevant conferences or meetings where possible. This review will inform future approaches to scale up antibiotic consumption surveillance strategies to attempt to maximise impact through standardisation.

**PROSPERO registration number** CRD42018103375

## Strengths and limitations of this study

► This study protocol follows the recommendations outlined in the Preferred Reporting Items for Systematic Review and Meta-analysis Protocols.
► This protocol has been registered on the International Prospective Register of Systematic Reviews.
► The selection of studies, data extraction and study quality assessments will be conducted by two independent reviewers.
► The exclusion of studies which do not report the total number of patients may exclude very large-scale antibiotic consumption surveillance projects.
► The information available in some published studies may be too limited to assess surveillance methodologies using the Affordability, Practicability, Effectiveness, Acceptability, Side-effects, Equity criteria.

population level higher rates of antibiotic resistance are associated with high levels of antibiotic consumption.[2] There are increasing data to indicate that antibiotic exposure promotes resistance at the individual level also.[3] Nationally aggregated antibiotic consumption across 65 countries is estimated to range between 4.4 and 64.4 defined daily doses (DDD) per 1000 inhabitants per day.[4] The Global Point Prevalence Survey estimated that 34.4% of adult inpatients admitted to 303 hospitals globally received at least one antimicrobial, and that 8.4% of adult inpatients were treated with a systemic antibacterial for a healthcare associated infection(s).[5] Antimicrobial stewardship (AMS) to optimise the use of antimicrobials and safely reduce the total quantity of antimicrobials consumed is a recognised approach to control the emergence and spread of AMR. Surveillance of antimicrobial consumption underpins the ability to monitor the impact of AMS interventions and to measure variation in appropriate/inappropriate antibiotic prescribing over time. Despite this, conventional metrics of antibiotic usage which

## INTRODUCTION

It is estimated that by the year 2050, 10 million deaths per year will be attributable to antimicrobial resistance (AMR).[1] There is good evidence from ecological studies that at the

enable comparisons across settings in secondary care have not been established.[6] In the currently published literature, there is also a lack of definitions or quantitative estimates of inappropriate antibiotic consumption in secondary care.

## Measuring antibiotic prescribing in secondary care

A range of strategies are being adopted to monitor and improve antibiotic prescribing in hospital. In Australia and England, surveillance systems have been established to measure antibiotic use in hospitals based on dispensing data from hospital pharmacies, converted into DDD.[7 8] A DDD is the assumed average maintenance dose per day for a drug for its main indication in adults.[9] DDDs are widely adopted to estimate quantities of antibiotic consumption and enable comparisons across drugs. However, the denominator of this measure in secondary care varies to make estimates less comparable, for example: 1000 occupied bed days, 1000 inhabitants, 1000 admissions.[10–13] This variation is likely to reflect differences in the availability of data across high-income secondary care settings. These metrics do not consider patient case-mix, nor is there is not a widely accepted methodology to apply to estimates to account for this. Consequently, any comparisons drawn between quantities of antibiotic usage across healthcare settings must be cautiously interpreted.

It is recommended that antimicrobial consumption surveillance should combine aggregate data with patient-level data to better understand patterns of prescribing.[4] In secondary care, patient-level data on prescribing and prescribing associated factors are usually obtained through point prevalence surveys, representing a single point in time.[14] However, secondary care is a dynamic setting and several clinicians may be involved in different stages of the prescribing process including: antibiotic initiation, treatment selection, clinical review, de-escalation and cessation of treatment. This contrasts with primary care, where patients usually have a single consultation which may result in an antibiotic prescription. This makes understanding patterns of prescribing within hospitals more difficult to map and highlights the usefulness of using patient-level prescribing data linked to the prescriber, to monitor patterns of antimicrobial usage and stewardship as patients move through the hospital.

## Measuring antibiotic stewardship

In the UK, the Commissioning for Quality and Innovation has established performance management targets to financially incentivise reductions in antibiotic use in primary and secondary care. This includes targets to promote prescribing review within 72 hours. In most hospitals, adherence to this target is monitored by auditing a sample of prescriptions.[15 16] This approach is time consuming and cannot capture the range of factors that might have an important bearing on the quality of the review, such as patient characteristics or the seniority or specialty of individuals who were involved in the prescribing process. Such factors limit the ability to make

inferences about the quality of AMS in different hospitals. Importantly existing measures provide little insight into where hospitals should target resources in their setting to have the greatest impact on antibiotic prescribing.

The National Antimicrobial Prescribing Survey (NAPS) measures similar antibiotic stewardship behaviours, however it goes further by estimating inappropriate prescribing. These data are collected through point prevalence surveys of all patients in the hospital where possible. Inappropriateness is defined in this survey as prescribing the wrong antibiotic spectrum, incorrect dose/frequency/duration/route and prescribing when an antimicrobial is not needed. Decisions on inappropriateness are made by a multi-desciplinary clinical team's assessment and are reported as prevalence of prescriptions. In 2017, 17 366 patients were submitted to NAPS with 26 227 prescriptions from 314 participating hospitals which were assessed for inappropriateness.[17] This is a large data collection workload which, in Australia, is only a proportion of the NAPS and runs annually alongside NAUSP. It is also a methodology which could be subject to variation between auditors when assessing inappropriateness of a broad range of prescriptions. Manual audits therefore seem labour-intensive and impractical for future attempts to scale up data collection of big, patient-level datasets to understand more granular patterns of prescribing in secondary care over time.

Variations exist in the length of time between data collection and publication of results across surveillance programmes. The 2016 NAUSP data were published in 2018, while the 2017 Swedish Antibiotic Utilisation and Resistance in Human Medicine (SWEDRES) and Swedish Veterinary Antibiotic Resistance Monitoring (SVARM) report published data collected in 2017 by the following year.[10 13] This likely reflects extensive differences in the availability of data, surveillance infrastructure and the digital maturity of healthcare systems which can be observed across high-income countries. The widespread establishment of electronic prescribing systems in the USA and the proposed investments in electronic prescribing systems in the UK and Australia, provides a potential opportunity to digitise systems to monitor antimicrobial usage and stewardship in secondary care and capture patient-level prescribing data over time to understand patterns of prescribing.[18 19] This could enable descriptions of patterns of prescribing within hospitals to better document appropriate and inappropriate antibiotic usage in secondary care, however it would rely upon systems which facilitate good documentation of stewardship behaviours and prescriber decision making.

As high-income settings answer the WHO call to increase the availability of antimicrobial use data, there is a need to reach a consensus on how to interrogate these data and be able to draw comparisons between hospital sites.[20] Previous attempts to establish a conventional approach have failed to address all of the issues which we have already highlighted. Recently, two studies attempted to establish a consensus on quality and quantity

indicators for measuring antibiotic use in secondary care using systematic review and RAND-modified Delphi methods.[21] [22] However, neither study considered differences in the availability of data and resources between hospital settings, the practicability of implementing these indicators in surveillance strategies, or the strengths and limitations of the measures. To develop robust systems to monitor antibiotic usage in secondary care requires the development of simple measures which enable comparisons of antibiotic usage over time and across specialties. Critically, these measures must be co-developed with clinicians and pharmacists, and be based on data that are routinely available, since few hospitals have the resources to conduct regular audits of antimicrobial use.

The Affordability, Practicability, Effectiveness and cost-effectiveness, Acceptability, Side-effects/safety and Equity (APEASE) criteria for designing interventions provides a framework within which researchers can comprehensively consider the context-specific suitability of an intervention.[23] It has been demonstrated repeatedly within the field in which it was developed to assess the suitability of behaviour change intervention functions.[24–27] It also provides the potential to be a useful tool when considering the complexities of identifying a suitable standardised antibiotic consumption surveillance methodology for the high-income secondary care setting.

This review will therefore study attempts to quantify total systemic antibiotic use, appropriate antibiotic use and inappropriate use in high-income settings in secondary care. This review will provide a summary of the ways in which antibiotic use can be measured alongside a narrative about the practicality of employing these metrics in secondary care based on an adaptation of the APEASE framework.

## OBJECTIVES
The aim of this review is to provide an outline of the methodologies to quantify antibiotic usage among inpatients in secondary care. The objectives are:
► To identify the data sources and metrics which have most frequently been used to represent antibiotic usage in secondary care.
► To assess the suitability of antibiotic use surveillance methodologies and metrics for use in routine surveillance of antibiotic usage based on the APEASE criteria.
► To document the variation in definitions and estimates of appropriate and inappropriate antibiotic usage.

## METHODS
The following methods are based on the framework provided by the Preferred Reporting Items for Systematic review and Meta-Analysis Protocols (PRISMA-P) checklist.[28] This review began in August 2018 and we aim to complete the review by September 2019.

## Eligibility criteria
### Types of studies
Study inclusion criteria:
► Studies which quantify antibiotic usage in at least 100 adult patients (aged >17 years) among inpatients in secondary care will be included.
► This review will include studies in the adult inpatient, high-income secondary care setting as it is expected that quantities of antibiotic use and patterns of prescribing in non-adult inpatients is likely to be significantly different to those in adults.
► Studies which focus on antibiotic usage among individuals with specific conditions or specific demographic groups will be included.
► Those studies which do not provide enough information for a complete assessment using the APEASE criteria will be included in the final review, but excluded from the APEASE criteria evaluation.
► This review will therefore include a range of study designs which are based on both observation and intervention, and which have attempted to estimate antibiotic usage in secondary care.
   Study exclusion criteria:
► This review will not include trials of antibiotic therapies in secondary care or reviews of the literature.
► Articles which describe a methodology for measuring antibiotic usage and which do not attempt to implement it to estimate antibiotic usage will be excluded.
► Articles which are not written in English language and are based on non-human participants will be excluded.
► If the number of adult inpatients included in the study is not stated, the article will be excluded from the review.

### Types of participants
The study must be based on dispensing, prescribing, survey/audit or sales data collected from the inpatient secondary care setting. It is expected that hospitals in high-income settings will be very different to those in low-resource settings in both the availability of data and resources available for stewardship, therefore this review will focus on hospitals in high-income countries as classified by the World Bank only (online supplementary table S1).[29] When the country is not classified in the World Bank lending groups, the article will be included if the gross national income per capita is US$12 056 or more. Articles must provide data which quantify antibiotic usage among adult patients (aged >17 years).

### Types of interventions
This review will consider all attempts to measure antibiotic usage among at least 100 patients in the hospital inpatient setting.

### Types of comparators
There are no comparison groups for this review as it is an exploration to identify how antibiotic usage has been

measured in secondary care and how this has been used to define and estimate appropriate and inappropriate antibiotic usage.

## Types of outcome measures

Articles which report a measure of antibiotic usage in hospital will be included in this review.

## Primary outcomes

► Data source used to represent antibiotic usage.
► Metrics which attempt to quantify antibiotic usage.
► An analysis of the suitability of existing metrics of antibiotic usage for use in routine practice to monitor antibiotic usage based on an adaptation of the APEASE framework (detailed in *Data Synthesis*).[23]

## Secondary outcomes

► Quantitative definitions of appropriate and inappropriate antibiotic usage.
► Comparison of the definitions of appropriate and inappropriate prescribing in secondary care.
► Description of variations in estimates of appropriate and inappropriate prescribing in secondary care.

## Information sources
### Electronic searches

The following databases will be searched for relevant articles:
► Medline (Ovid);
► Embase (Ovid);
► Cumulative Index to Nursing and Allied Health Literature;
► Cochrane Central Register of Controlled Trials.

The search strategy will be developed in Medline and then adapted appropriately for use in each database. The search will be conducted across all databases on 2 August 2018.

### Website searches

In order to capture reports from the grey literature, the websites of relevant organisations will be searched (online supplementary table S2). The list of organisations expands on a list originally conceived by Stanić Benić *et al.* based on discussion between SP, AJ and LS.[22] This review aims to consider the suitability of existing metrics of antibiotic usage in routine hospital surveillance of antibiotic usage; therefore it is essential that approaches to do this on a large scale such as through National Action Plans, which are often published in the grey literature, should be captured in our search. This search will be conducted in February 2019, the specific dates of which will be reported in the final review.

### Search strategy

The provisional search strategy is adapted from the work of Stanić Benić *et al.* based on four concepts: antibiotics, utilisation, measure, hospital (table 1).[22] The search strategy will be established in Medline and then adapted to the other three databases included in the search. No additional restrictions will be placed on the search to maximise sensitivity.

## Study records
### Data management

The articles will be uploaded from the relevant database to *Mendeley* reference manager software. The reports will be exported to the web-based, *Evidence for Policy and Practice Information (EPPI) Reviewer 4* systematic review software which will be used to remove duplicate articles. Each stage of the screening process will be carried out in *EPPI Reviewer 4* using a standardised form of article eligibility. The online form for article eligibility will be piloted before the start of the screening process and amended as required to ensure it is fit for purpose. The data will be extracted and entered into a standardised, piloted, precreated online data extraction form in *EPPI Reviewer 4* in duplicate by two independent reviewers (SP and AJ).

### Selection process

The unique articles captured by the search will undergo a title prescreening by a single reviewer (SP). The remaining articles will then be screened based on title and abstract by two independent reviewers in duplicate (SP and AJ); when study eligibility is not mutually agreed on or is ambiguous from the title and abstract only, the article will be admitted into the full-text review stage. The selected articles will undergo full-text review by two independent reviewers in duplicate (SP and AJ). Any disagreements between authors at any stage will be resolved through a third reviewer (LS). The reasons for the exclusion of these studies from this review will be documented for reference and the study selection process will be presented as a PRISMA flow diagram.[28]

### Data collection process

The data will be collected in duplicate by two independent reviewers (SP and AJ) from articles selected for inclusion in the review. The data will be entered into precreated, piloted forms on EPPI Reviewer 4: systematic review software. Any disagreement will be resolved through consultation with a third author (LS).

### Data items

We will extract data on:
► *Study* (study design, country, population, service setting such as size of hospital, objective).
► *Data type used to describe antibiotic usage* (data source, method of collection, frequency of the data collection, reason for data collection eg. national surveillance or research, who collected the data, level of detail in the data and data type).
► *Metrics used to measure antibiotic usage* (numerator, denominator, units (such as per admission), level at which the metric is presented (such as by specialty).
► *Criteria used to define appropriate and inappropriate antibiotic usage.*

**Table 1** Medline (Ovid) provisional search terms

| Search concept | Search terms |
|---|---|
| Antibiotic | 1. Antibacterial agents/ad, dt, sd, tu, th, ut (Administration & Dosage, Drug Therapy, Supply & Distribution, Therapeutic Use, Therapy, Utilization) |
| | 2. Antibiotic prophylaxis/ec, mt, sn, td, ut (Economics, Methods, Statistics & Numerical Data, Trends, Utilization) |
| | 3. (anti?biotic? or anti?microbial? or anti?bacterial?).ab,ti. |
| | 4. 1 or 2 or 3 |
| Utilisation | 5. Drug prescriptions/ |
| | 6. Drug utilization/ |
| | 7. 'Drug utilization review'/cl, ec, mt, st, sn, td, ut (Classification, Economics, Methods, Standards, Statistics & Numerical Data, Trends, Utilization) |
| | 8. ((anti?biotic? or anti?microbial?) adj3 (prescri* or consumption or utili?ation or usage or 'use' or dispens* or sale?)).ab,ti. |
| | 9. 5 or 6 or 7 or 8 |
| Measurement | 10. BENCHMARKING/cl, ec, mt, st, sn, td, ut (Classification, Economics, Methods, Standards, Statistics & Numerical Data, Trends, Utilization) |
| | 11. (intervention adj5 (prescri* or stewardship or 'use' or utili?ation or usage or consumption)).ab,ti |
| | 12. ((anti?biotic? or anti?microbial?) adj4 (estimat* or quanitf* or metric? or monitor* or surveillance or prevalence or survey or audit)).ab,ti. |
| | 13. (electronic prescri* or e?prescri*).ab,ti. |
| | 14. 10 or 11 or 12 or 13 |
| Secondary care | 15. Secondary care/ |
| | 16. Hospitals/ |
| | 17. hospital*.ab,ti. |
| | 18. 15 or 16 or 17 |
| | 19. 4 and 9 and 14 and 18 |

ab, abstract; ti., title; adj3–5, indicates two words next to each other in any order with up to 2–4 words in between; /, indicates a Medical Subject Heading (MeSH); *, denotes any truncation; ?, denotes one character or no character. All other abbreviations are elaborated in parentheses in the table.

► *Data type used to quantify appropriate and inappropriate antibiotic usage* (data source, method of collection, level of detail in the data and data type).

► *Quantitative estimates of appropriate and inappropriate antibiotic usage* (numerator, denominator, units (such as per admission), level at which the metric is presented (such as by specialty) and what the metric is currently used for (such as national or hospital-level surveillance).

► *Study quality and selection bias of those studies which provide an estimate of appropriate and inappropriate antibiotic usage* (outlined under 'Assessment of risk of bias individual studies' section).

### Outcomes prioritisation

The main outcome of this review is a description of the data sources and metrics which have been used to measure antibiotic usage and to define appropriate

**Table 2** An adaptation of the APEASE framework to assess the feasibility of metrics for use in routine surveillance

| | | |
|---|---|---|
| A | (Affordability) Resources required to sustainably implement the metric. For example, person time to audit/new system roll-out. |
| P | (Practicability) Can the metric be delivered? Is there sustainable access to data and resources? |
| E | (Effectiveness) Does the surveillance method capture sufficient data? |
| A | (Acceptability) Does it represent relevant data to monitor antibiotic usage and appropriateness of antibiotic usage? |
| S | (Side-effects) Bias of metrics |
| E | (Equity) Is it feasible for hospital trusts of varying levels of resources and digital maturity to all implement the surveillance methodology? |

 

and inappropriate antibiotic usage in hospitals. This will include a description of the teams most frequently involved in this type of work in order to identify those groups who are often facilitators of this surveillance.

This will then be used to consider each surveillance method within an adaptation of the APEASE framework to assess the practicality of each method and metric for use in routine surveillance of antibiotic usage; this method is detailed in 'Data synthesis'.[23] This aims to inform the methodology of sustainable antibiotic usage surveillance programmes. We will also document if it is stated that the metric is currently being routinely used for any purpose such as national level surveillance. This will identify reasons why antibiotic usage surveillance is already being implemented and combined with methodological details will reveal programmes where existing infrastructure may be extended for use to routinely monitor antibiotic usage within hospitals. If the metric has been used to define and estimate (in)appropriate antibiotic usage, this will be used to explore the variation in the definitions and estimates of inappropriate antibiotic usage in hospitals. It is anticipated that such estimations will have been made using a multitude of methodologies and measures. The extent to which quantitative estimates of inappropriate prescribing can be directly compared may therefore be limited.

### Assessment of risk of bias in individual studies

All metrics of antibiotic consumption included in the final review must have been used in the secondary care setting to capture the antibiotic usage data for at least 100 patients. This is to ensure some level of demonstration of the feasibility of implementing the measures. For these descriptive purposes, it is not appropriate to use a risk of bias framework.

For studies included in the full-text review stage and which attempt to make a quantitative estimate of appropriate or inappropriate antibiotic usage in the secondary care setting, the Newcastle-Ottawa scale to assess study quality will be used to determine the inclusion of the estimate in the review.[30]

### Data synthesis

The data will undergo a descriptive synthesis to provide insight into the methodologies to implement antibiotic usage metrics in secondary care. We will record the frequency of use of different data sources by level (individual to national) and by source (such as electronic prescribing or audits). This will examine the type of data which is currently being used to monitor antibiotic usage and consequently the level detail in which we are able to examine prescribing currently. We will also identify the frequency of use of each metric to measure: volume, type, duration, stewardship activity and prevalence/incidence of antibiotic use. As well as identify the level at which these metrics are most frequently presented such as by specialty or nationally. The teams involved in these studies to collect and analyse the data will also be recorded.

An adaptation of the APEASE framework will be used to assess the feasibility of metrics for use in routine monitoring of antibiotic usage. The APEASE framework was designed to aid the context-specific design and selection of interventions to promote health behaviour change.[23] This original framework will be adapted to inform the establishment of a consensus to monitor antibiotic usage in the high-income secondary care setting (table 2).

This analysis can then be used to inform recommendations based on the available literature regarding a sustainable, conventional strategy to monitor antibiotic usage and the resources which would be required to practically implement such a surveillance strategy.

The variation between comparable hospital-level estimates of (in)appropriate antibiotic usage will be represented using a funnel plot of inappropriate prescribing estimates against patient numbers and 95% CIs.

### Meta-bias(es)

The systematic review will be conducted in line with this protocol and any aspects of the method which deviate from the original protocol will be reported in the final review. Only articles written in English language will be included in this review; this is due to the language spoken by the authors.

### Confidence in cumulative evidence

This is a descriptive review to consider the methodologies which have been used to measure antibiotic usage and to define appropriate and inappropriate antibiotic prescribing in hospitals. The methodologies will be considered in terms of the APEASE criteria and it is therefore not necessary to consider the quality of quantitative evidence for all studies. For observational studies which attempt to make a quantitative estimate of appropriate or inappropriate antibiotic usage in the secondary care setting, the Newcastle-Ottawa scale to assess study quality will be used to determine the inclusion of the estimate in the review.[30]

**Acknowledgements** The authors acknowledge the valuable technical guidance offered by Suvi Härmälä in the construction of the protocol.

**Contributors** The study was conceived by SP, SH and LS. SP developed the eligibility criteria, search strategy, risk of bias assessment and the data extraction plans with guidance from LS. SP, AJ and LS developed the website search list. All authors contributed to, reviewed and approved the final manuscript.

**Funding** This review was supported by the Economic and Social Research Council (ESRC) [grant number ES/P000592/1 awarded to SP].

**Disclaimer** The ESRC was not involved in the development of this protocol.

**Competing interests** None declared.

**Patient consent for publication** Not required.

**Provenance and peer review** Not commissioned; externally peer reviewed.

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
