## [Reviewer comments · BMJ Open]

ARTICLE DETAILS

TITLE (PROVISIONAL)	HOW CAN METRICS BE USED TO ROUTINELY MONITOR ANTIBIOTIC USAGE AND DEFINE (IN)APPROPRIATE ANTIBIOTIC USAGE IN SECONDARY CARE? A SYSTEMATIC REVIEW PROTOCOL
AUTHORS	Patel, Selina; Jhass, Arnoupe; Hopkins, Susan; Shallcross, Laura

VERSION 1 - REVIEW

REVIEWER	Kathleen Holloway Institute of Development Studies, University of Sussex, Brighton, UK
REVIEW RETURNED	13-Nov-2018

GENERAL COMMENTS	This manuscript describes a systematic review protocol for how metrics can be used to routinely monitor antibiotic usage and define (in)appropriate antibiotic usage in secondary care. This review will be very useful since there is currently no agreed set of indicators or methods for routine monitoring of (in)appropriate antibiotic usage in secondary inpatient care - that allow comparison between regions and countries. The protocol is well written but would benefit from inclusion of a few more actual examples of indicators and methods already used in the literature, together with their limitations, so that the reader may better understand what this review will add to previous reviews. Also the APEASE criteria should be explained or reference made to table 2 when it is first mentioned in the text. Another limitation is that many published articles describing metrics of antibiotic usage may not sufficiently describe the context and therefore it may not be possible to assess all metrics on the basis of the APEASE criteria. How will this be dealt with in the review? Will these studies be included or excluded? Specific comments are as follows. Abstract: Mention is made of the Global Point Prevalence Survey in the abstract but not in the introduction and no reference given. This sentence should be incorporated in the introduction and reference given. Introduction: It is mentioned in the penultimate sentence that digitized descriptions of prescribing patterns within hospitals would enable better understanding of the drivers of inappropriate antibiotic usage, but this may not be so, as much information will be needed on context and stewardship activities. This is why the APEASE criteria are being used, so this sentence needs re-phrasing.
--

	Objectives: The APEASE criteria should perhaps be mentioned under the second objective since they are the main way of assessing the suitability of indicators and methods for routine surveillance - which is the main reason for doing this review. Methods Types of participants: why are children under the age of 18 excluded? Many inpatients are children. Primary outcomes: please mention table 2 when mentioning the APEASE framework and provide a web address. Information sources: please write out CINAHL in full. Table 1: please explain all the acronyms in a legend at the bottom of the table. Some acronyms could not be deciphered from the table e.g. "ti", "adj". What is the meaning of "?" or "*" in the middle of search terms? Outcome Prioritisation: why will documentation that a metric is already being used for national surveillance identify reasons why antibiotic surveillance is already being implemented routinely? Other contextual information may be needed to understand why surveillance is being done routinely and what extra infrastructure may be needed to extend surveillance to other hospitals. Assessment of risk of bias: It is mentioned that where the number of patients for which data has been captured using these metrics has not been detailed, the article will be included. This is likely to result in the inclusion of metrics measured in less than 100 patients contrary to the protocol methodology since poor methodology and small samples sizes are often not highlighted in articles. Confidence in cumulative evidence: it is mentioned that it is not necessary to consider the quality of quantitative evidence since the review is descriptive only of methodologies used to measure antibiotic usage. Surely, poor methodologies may be reflected by poor quality quantitative evidence. Furthermore, objective 3 is a description of variations in estimates of (in)appropriate antibiotic usage and this will be affected by poor methods and poor quality evidence. Data synthesis, last sentence: please expand on how the variation of between comparable hospital-level estimates of (in)appropriate antibiotic usage will be analysed. Table S2: the National Prescribing Service in Australia, though mentioned in the introduction, is not included in website search list but may have important information. References: please provide a web address for the APEASE framework.
--	--

REVIEWER	Sajal Saha Monash University Australia
REVIEW RETURNED	19-Nov-2018

GENERAL COMMENTS	This systematic review protocol is generally well written. However, I have some comments. 1. Abstract. The introduction of the abstract could be shorter. APEASE term could be spelled out. Risk of bias assessment tools could be mentioned. Main outcomes should be clearly written points by points. One sentence can be added on the implication of anticipated findings at the end.
--

	2. Strength and limitation could be re-written and some points could be more specific to the research topics, findings, and its implications 3. The introduction could considerably be shorter. Much of the information on the first paragraph is so well-known, this could be described concisely. Yet, an attention could be paid to the global estimates of antibiotic usage, secondary care specific antibiotic consumption and appropriateness data (if any) with current references and usage of metrics, why it is so important. There is information in the introduction but I miss a clear definition of metrics and how these are used. Defining metrics and its importance may help increase readability and clarity to the reader. APEASE framework can be described as a bit to familiarize prior to the last paragraph. 4. In objective two, how would the author assess the suitability of metrics (if by using APEASE framework or others) then state? 5. Types of study. The author will quantify antibiotic usage in at least 100 adult patients (>17 years old) among in-patients in secondary care. However, what is the rationale for restricting patients >17 years old? Need explanation. 6. Inclusion criteria for studies is not clear. Inclusion and exclusion criteria can be written separately points by points for clarity. This sentences, "Where the number of patients is not detailed, the article will be excluded from the review" can be modified. 6. Types of Participants The last sentence is confusing in terms of patient age groups (inclusion or exclusion). 7. Outcomes are well mentioned. However, prior definitions of appropriate and inappropriate antibiotic use would help compare the variation in reported studies. I wonder how the author would like to deal with the contextual factors which might have influence on appropriate and inappropriate use of antibiotics. Appropriateness and inappropriateness of antibiotic use are sometimes context dependent. 8. Database search. It is better to mention in the text the time period of your search for each database. 9. I found no information on how could you deal with the sensitivity and specificity of your database search? 10. The dates of the study need to be included in the manuscript
--	--

VERSION 1 – AUTHOR RESPONSE

Reviewers' Comments to Author:

Reviewer: 1

Reviewer Name: Kathleen Holloway

Institution and Country: Institute of Development Studies, University of Sussex, Brighton, UK

Please state any competing interests or state 'None declared': None declared.

This manuscript describes a systematic review protocol for how metrics can be used to routinely monitor antibiotic usage and define (in)appropriate antibiotic usage in secondary care. This review will be very useful since there is currently no agreed set of indicators or methods for routine monitoring of (in)appropriate antibiotic usage in secondary inpatient care - that allow comparison between regions and countries.

The protocol is well written but would benefit from inclusion of a few more actual examples of indicators and methods already used in the literature, together with their limitations, so that the reader may better understand what this review will add to previous reviews. Now included in the Introduction.

Also the APEASE criteria should be explained or reference made to table 2 when it is first mentioned in the text. Now included towards the end of the Introduction.

Another limitation is that many published articles describing metrics of antibiotic usage may not sufficiently describe the context and therefore it may not be possible to assess all metrics on the basis of the APEASE criteria. How will this be dealt with in the review? Will these studies be included or excluded? Addressed in Types of Studies / Exclusion Criteria.

Specific comments are as follows.

Abstract:

Mention is made of the Global Point Prevalence Survey in the abstract but not in the introduction and no reference given. This sentence should be incorporated in the introduction and reference given. This reference has now been included in the Introduction.

Introduction:

It is mentioned in the penultimate sentence that digitized descriptions of prescribing patterns within hospitals would enable better understanding of the drivers of inappropriate antibiotic usage, but this may not be so, as much information will be needed on context and stewardship activities. This is why the APEASE criteria are being used, so this sentence needs re-phrasing. Amended in 3rd to last paragraph of the Introduction.

Objectives:

The APEASE criteria should perhaps be mentioned under the second objective since they are the main way of assessing the suitability of indicators and methods for routine surveillance - which is the main reason for doing this review. Method now stated in Objective 2.

Methods

Types of participants: why are children under the age of 18 excluded? Many inpatients are children. Now stated in Types of Studies.

Primary outcomes: please mention table 2 when mentioning the APEASE framework and provide a web address. The data synthesis chapter has been referenced objective 2 which details our adaptation of the APEASE criteria, in lieu of referencing Table 2. This avoids incorrect ordering of table citations. The authors have not found a web address reference for the APEASE framework; the relevant book reference has been included.

Information sources: please write out CINAHL in full. Full title now included in Electronic searches.

Table 1: please explain all the acronyms in a legend at the bottom of the table. Some acronyms could not be deciphered from the table e.g. "ti", "adj". What is the meaning of "?" or "*" in the middle of search terms? Search Terms table has been updated to include a legend.

Outcome Prioritisation: why will documentation that a metric is already being used for national surveillance identify reasons why antibiotic surveillance is already being implemented routinely? Other contextual information may be needed to understand why surveillance is being done routinely and what extra infrastructure may be needed to extend surveillance to other hospitals. This sentence in the Outcomes Prioritisation section has been restructured.

Assessment of risk of bias: It is mentioned that where the number of patients for which data has been captured using these metrics has not been detailed, the article will be included. This is likely to result in the inclusion of metrics measured in less than 100 patients contrary to the protocol methodology since poor methodology and small samples sizes are often not highlighted in articles. If the number of patients included in the study is not reported, the study will be excluded from the final review.

Confidence in cumulative evidence: it is mentioned that it is not necessary to consider the quality of quantitative evidence since the review is descriptive only of methodologies used to measure antibiotic usage. Surely, poor methodologies may be reflected by poor quality quantitative evidence. Furthermore, objective 3 is a description of variations in estimates of (in)appropriate antibiotic usage and this will be affected by poor methods and poor quality evidence. Further detail has been added to the confidence in cumulative evidence section of the protocol for clarification. The descriptive part of this review is to assess the context-specific suitability of antibiotic consumption surveillance methods using the APEASE criteria whilst the quantitative element of the review will undergo an assessment using the Newcastle-Ottawa scale for observational studies.

Data synthesis, last sentence: please expand on how the variation of between comparable hospital-level estimates of (in)appropriate antibiotic usage will be analysed. Further detail has been added to the final sentence in Data Synthesis.

Table S2: the National Prescribing Service in Australia, though mentioned in the introduction, is not included in website search list but may have important information. Now included in Table S2, number 5.

References: please provide a web address for the APEASE framework. The authors have been unable to find a web address reference for the APEASE framework. The Behaviour Change Wheel Book which details this methodology has been referenced.

Reviewer: 2

Reviewer Name: Sajal Saha

Institution and Country: Monash University, Australia

Please state any competing interests or state 'None declared': None

This systematic review protocol is generally well written. However, I have some comments.

1. Abstract. The introduction of the abstract could be shorter. APEASE term could be spelled out. Risk of bias assessment tools could be mentioned. Main outcomes should be clearly written points by points. One sentence can be added on the implication of anticipated findings at the end. Abstract Introduction has been made slightly more concise. APEASE term has been expanded. Outcomes have been stated sentence by sentence in Abstract Methods and Analysis. Implication of anticipated findings sentence added to Abstract Ethics and Dissemination.

2. Strength and limitation could be re-written and some points could be more specific to the research topics, findings, and its implications. Limitations added.

3. The introduction could considerably be shorter. Much of the information on the first paragraph is so well-known, this could be described concisely. Yet, an attention could be paid to the global estimates of antibiotic usage, secondary care specific antibiotic consumption and appropriateness data (if any) with current references and usage of metrics, why it is so important. There is information in the introduction but I miss a clear definition of metrics and how these are used. Defining metrics and its importance may help increase readability and clarity to the reader. APEASE framework can be described as a bit to familiarize prior to the last paragraph. More detailed examples of measures and

existing surveillance programmes and their limitations are now included in the Introduction. The APEASE framework is now also described in the 2nd to last paragraph of the Introduction. As a consequence, the length of the introduction is no shorter.

4. In objective two, how would the author assess the suitability of metrics (if by using APEASE framework or others) then state? Method now stated in Objective 2.

5. Types of study. The author will quantify antibiotic usage in at least 100 adult patients (>17 years old) among in-patients in secondary care. However, what is the rationale for restricting patients >17 years old? Need explanation. Now stated in Types of Studies.

6. Inclusion criteria for studies is not clear. Inclusion and exclusion criteria can be written separately points by points for clarity. This sentences, "Where the number of patients is not detailed, the article will be excluded from the review" can be modified. This Types of Studies section and the highlighted sentence has been amended for clarity.

6. Types of Participants

The last sentence is confusing in terms of patient age groups (inclusion or exclusion). This sentence has been rephrased in Types of Participants.

7. Outcomes are well mentioned. However, prior definitions of appropriate and inappropriate antibiotic use would help compare the variation in reported studies. I wonder how the author would like to deal with the contextual factors which might have influence on appropriate and inappropriate use of antibiotics. Appropriateness and inappropriateness of antibiotic use are sometimes context dependent. The third objective of the review in the Objectives section is to document the definitions and estimates of appropriate and inappropriate prescribing. The definition of these two things will consequently be data driven and not decided by the authors. If definitions vary enough across published studies that estimates are not comparable, this will represent a limitation in the current published literature which would need to be explored further in the final review.

8. Database search. It is better to mention in the text the time period of your search for each database. Now included in Electronic searches.

9. I found no information on how could you deal with the sensitivity and specificity of your database search? Now referenced in Search Strategy.

10. The dates of the study need to be included in the manuscript. Review dates are reported in the first paragraph of Methods and Information Sources.

VERSION 2 – REVIEW

REVIEWER	Kathleen Holloway Institute of Development Studies, Sussex University, UK
REVIEW RETURNED	19-Feb-2019

GENERAL COMMENTS	This article is much improved and the authors have addressed the reviewer comments. A few sentences could still be clarified. Introduction, page 6, last paragraph: digitizing prescribing will help to identify types of antibiotic misuse, but not necessarily drivers of drivers of misuse which may require consideration of other health system and staff factors, which in turn may require qualitative investigation.
--

	Methods, page 9, 2nd bullet: exclusion of studies that do not provide enough APEASE information will be excluded from WHICH stage of the analysis? The quantitative stage? Methods, page 15, lines 29-31: documentation of whether a metric is currently being used routinely will not of itself identify motivations and reasons for why surveillance is being done but rather may allow identification of such motivations and reasons when analysed in conjunction with the APEASE criteria. Methods, page 18, 1st paragraph: is it meant that "For observational studies....., the Ottawa-Newcastle scale will be used... "?
--	---

REVIEWER	Sajal K. Saha Monash University, Australia and the University of Dhaka, Bangladesh
REVIEW RETURNED	03-Mar-2019

GENERAL COMMENTS	The revised version of the manuscript has been amended satisfactorily. However, addressing a few issues can improve readability. Abstract The key objective of this review should be included as the last sentence of the introduction section of the abstract. Methods section Inclusion criteria First point-i) Start with the sentence like "The article that quantifies antibiotic usage....." Divide the second sentence as a point of inclusion criteria like ii) the study that focuses on adult inpatient of high-income country settings 3rd point- "this stage of analysis" this would be replaced by "the stage of analysis". 4th point- Directly state that this study will allow a range of study design..... Delete "for a variety of purposes". Study Records section EPPI can be spelled out when first written
--

VERSION 2 – AUTHOR RESPONSE

Reviewer(s)' Comments to Author:

Reviewer: 1

Reviewer Name: Kathleen Holloway

Institution and Country: Institute of Development Studies, Sussex University, UK

Please state any competing interests or state 'None declared': None declared

Please leave your comments for the authors below

This article is much improved and the authors have addressed the reviewer comments. A few sentences could still be clarified.

Introduction, page 6, last paragraph: digitizing prescribing will help to identify types of antibiotic misuse, but not necessarily drivers of drivers of misuse which may require consideration of other health system and staff factors, which in turn may require qualitative investigation. This sentence has been amended.

Methods, page 9, 2nd bullet: exclusion of studies that do not provide enough APEASE information will be excluded from WHICH stage of the analysis? The quantitative stage? This sentence has been clarified.

Methods, page 15, lines 29-31: documentation of whether a metric is currently being used routinely will not of itself identify motivations and reasons for why surveillance is being done but rather may allow identification of such motivations and reasons when analysed in conjunction with the APEASE criteria. This sentence has been made more concise.

Methods, page 18, 1st paragraph: is it meant that "For observational studies...., the Ottawa-Newcastle scale will be used..."? This is correct. This sentence has now been corrected.

Reviewer: 2

Reviewer Name: Sajal K. Saha

Institution and Country: Monash University, Australia

and the University of Dhaka, Bangladesh

Please state any competing interests or state 'None declared': None.

Please leave your comments for the authors below

The revised version of the manuscript has been amended satisfactorily. However, addressing a few issues can improve readability.

Abstract

The key objective of this review should be included as the last sentence of the introduction section of the abstract. This has been added. We have made minor additional alterations to the abstract to satisfy the word limit following this amendment.

Methods section

Inclusion criteria

First point-i) Start with the sentence like "The article that quantifies antibiotic usage....." The start of this sentence has been simplified.

Divide the second sentence as a point of inclusion criteria like ii) the study that focuses on adult inpatient of high-income country settings This has been altered as suggested.

3rd point- "this stage of analysis" this would be replaced by "the stage of analysis". This point has been clarified.

4th point- Directly state that this study will allow a range of study design.....

Delete "for a variety of purposes". This has been deleted and ammended.

Study Records section

EPPI can be spelled out when first written. This has been expanded.